# The Effectiveness of Interventions to Evaluate and Reduce Healthcare Costs of Potentially Inappropriate Prescriptions among the Older Adults: A Systematic Review

**DOI:** 10.3390/ijerph19116724

**Published:** 2022-05-31

**Authors:** Sara Mucherino, Manuela Casula, Federica Galimberti, Ilaria Guarino, Elena Olmastroni, Elena Tragni, Valentina Orlando, Enrica Menditto

**Affiliations:** 1CIRFF—Center of Pharmacoeconomics and Drug Utilization, Department of Pharmacy, Federico II University of Naples, 80131 Naples, Italy; sara.mucherino@unina.it (S.M.); ilaria.guarino@cirff.it (I.G.); valentina.orlando@unina.it (V.O.); 2Epidemiology and Preventive Pharmacology Service (SEFAP), Department of Pharmacological and Biomolecular Sciences, University of Milan, 20133 Milan, Italy; manuela.casula@unimi.it (M.C.); elena.olmastroni@unimi.it (E.O.); elena.tragni@unimi.it (E.T.); 3IRCCS MultiMedica Hospital, Sesto S. Giovanni, 20099 Milan, Italy; federica.galimberti@multimedica.it

**Keywords:** potentially inappropriate prescribing, healthcare costs, educational interventions

## Abstract

Potentially inappropriate prescribing (PIP) is associated with an increased risk of adverse drug reactions, recognized as a determinant of adherence and increased healthcare costs. The study’s aim was to explore and compare the results of interventions to reduce PIP and its impact on avoidable healthcare costs. A systematic literature review was conducted according to Preferred Reporting Items for Systematic reviews and Meta-Analyses (PRISMA) statement guidelines. PubMed and Embase were queried until February 2021. Inclusion criteria followed the PICO model: older patients receiving PIP; Interventions aimed at health professionals, structures, and patients; no/any intervention as a comparator; postintervention costs variations as outcomes. The search strategy produced 274 potentially relevant publications, of which 18 articles met inclusion criteria. Two subgroups were analyzed according to the study design: observational studies assessing PIP frequency and related-avoidable costs (*n* = 10) and trials, including specific intervention and related outcomes in terms of postintervention effectiveness and avoided costs (*n* = 8). PIP prevalence ranged from 21 to 79%. Few educational interventions carried out to reduce PIP prevalence and avoidable costs resulted in a slowly improving prescribing practice but not cost effective. Implementing cost-effective strategies for reducing PIP and clinical and economic implications is fundamental to reducing health systems’ PIP burden.

## 1. Introduction

Population aging is occurring along with broader social and economic changes that are taking place around the world [1]. According to the World Health Organization (WHO), between 2015 and 2050, the proportion of the world’s population over 60 years will nearly double from 12 to 22%. This demographic transition has a number of implications for healthcare in terms of both health outcomes and healthcare costs, as older adults are prone to multiple chronic conditions [2,3], necessitating the use of multiple medications or polypharmacy [4,5]. The increase in age and consequently in the patient clinical complexity certainly are factors that can hinder the rational use of medicines. The WHO defined the rational use of medicines as a situation where “patients receive medications appropriate to their clinical needs, in doses that meet their own individual requirements, for an adequate period of time, and at the lowest cost to them and their community” [6]. Rational prescribing refers to a process that emphasizes how prescribing decisions are to be made. Although this issue has multifactorial reasons and causes, the lack of appropriate dialogue between doctors and patients, and doctors’ competency concerns, may result in inappropriate prescribing; prescribers may lack important information justifying the prescription of a particular drug or therapy [7]. Thus, prescriptive inappropriateness can result from multiple patient-related reasons, such as seeking nonmedical, complementary, and alternative treatment, not in agreement with the prescriber. Another risk factor for prescriptive inappropriateness is age. In fact, with advancing age, the complexity of the patient’s clinical picture increases, which can result in inappropriate polypharmacy and poor adherence to drug therapies.

Appropriateness in healthcare has already been described as the outcome of a decision-making process that maximizes net individual health gains within society’s available resources [8]. Potentially inappropriate prescribing (PIP) is a phenomenon related to an increased risk of adverse drug reactions (ADRs) as well as increased healthcare costs.

In recent years, many strategies and tools have been developed to assess the appropriateness of medication use by developing lists of explicit criteria to identify potentially inappropriate medication (PIM) use, especially among older people [9,10,11,12,13,14,15,16,17,18]. The use of these criteria so far has allowed PIP to be assessed, detecting high levels among adults [9,10,11,12,13,14,15,16,17,18]. Among the criteria most used to date are the Beers Criteria for Potentially Inappropriate Medication Use in Older Adults, commonly called the Beers List, which is a guideline to improve prescribing appropriateness in older adults (65 years and older); the PRISCUS list, firstly created for the German pharmaceuticals market and also developed to improve prescribing appropriateness in older adults; STOPP (Screening Tool of Older Persons’ Prescriptions) and START (Screening Tool to Alert to Right Treatment) criteria, developed for clinicians and aimed to review prescription inappropriateness in older adults.

Likewise, in many countries and different healthcare settings, educational interventions addressed to prescribers, pharmacists, geriatricians, and other health professionals have been implemented in order to monitor PIMs and try to reduce the phenomenon as a determinant of ADRs and increased healthcare costs [19,20,21,22,23,24,25,26,27,28].

In this scenario, the Italian Medicine Agency (Agenzia Italiana del Farmaco—AIFA) funded the EDU.RE.DRUG Project (“Effectiveness of informative and/or educational interventions aimed at improving the appropriate use of drugs designed for general practitioners and their patients”). The EDU.RE.DRUG Project aims to evaluate the appropriateness of drug prescription among older adults living in Northern and Southern Italy [29,30]. Therefore, the aim of this review was to explore PIP-related avoidable costs and the results of any interventions to reduce PIP among the older adult population, and the pertaining impact on healthcare costs.

## 2. Materials and Methods

### 2.1. Information Sources

A systematic review of the literature was conducted according to the PRISMA (Preferred Reporting Items for Systematic reviews and Meta-Analyses) statement guidelines [31].

### 2.2. Search Strategy

A systematic search of the published peer-reviewed literature was carried out without time limits. The identification of relevant studies was achieved by searching electronic databases of the published literature, including the Medical Literature Analysis and Retrieval System Online (via PubMed/MEDLINE) and Embase (via Ovid). Although MEDLINE was chosen as the source as the main bibliographic database containing more than 29 million references to journal articles in the life sciences, PubMed/MEDLINE does not fully cover the research literature in the field. For this reason, Embase was used as a supplementary source to Medline/PubMed to obtain a comprehensive overview of the papers in the field of potentially inappropriate prescribing and related avoidable costs.

First, the search strategy was developed and completed in PubMed/Medline, and then the same strategy was applied to Embase (excluding Medline literature). More in detail, the search strategy combined headings and keywords identified according to the PICO Model. Literature search strategies were developed using Medical Subject Headings (MeSH) terms when performed in PubMed or Excerpta Medica Thesaurus (Emtree terms) when performed in Embase and included titles and abstracts.

To reach the systematic review objective, the strategy combined four major themes with their synonyms: (i) inappropriate prescribing of medications; (ii) older adult population; (iii) cost evaluation; (iv) intervention on health professionals or patients. The Boolean operators used were AND/OR. The full search strategy performed is reported in Table 1. The search syntax is presented in detail in Appendix A.

### 2.3. Eligibility Criteria

The inclusion criteria were based on compliance with the PICO model, as follows:Patient (P): subjects receiving any specific and nonspecific PIP, aged 65 and over;Intervention (I): Any type of intervention aimed at health professionals (i.e., physicians, clinicians, pharmacists, nurses), structures (i.e., nursing homes, hospitals, pharmacies), patients, or any monitoring activities of potentially inappropriate prescriptions and costs incurred, using any explicit criteria for identifying inappropriateness (such as Beers Criteria, PRISCUS list, STOPP/START criteria);Comparator (C): No intervention or any other intervention and no monitoring activities;Outcomes (O): Postintervention outcomes in terms of cost variation, intervention effectiveness, and avoidable healthcare costs.

All studies responding to the PICO were included in the research. Hence, peer-reviewed original articles published in any time frame up to February 2021 were included. Particularly, all studies about reporting any intervention aimed to investigate or improve PIP in older adult patients and how this improvement may impact avoidable healthcare costs were included (inclusion criteria).

On the other hand, conference proceedings, rationale or design, letters, editorials, commentaries, reviews, consensus, and study protocol were not included (exclusion criteria). Moreover, no language restriction was applied to the first step of research, but fundamental to the eligibility of the study was the availability of the papers’ full text published in English.

### 2.4. Selection and Data Process

The references were collected using the software program Endnote, version X9 (Clarivate Analytics, Philadelphia, PA, USA). All references were screened for relevance, and those potentially eligible were assessed according to the inclusion/exclusion criteria, accepted or rejected, as appropriate.

Four researchers screened titles and abstracts in pairs in a double-blind method to discard irrelevant ones in the first screening phase. Then, the four researchers together assessed full texts for eligibility, defining which references to include in the qualitative analysis. The references obtained were validated by four other expert researchers in the fields of pharmacology, drug utilization, and pharmacoeconomics. Full texts of relevant studies were reviewed for eligibility in accordance with the inclusion criteria. The risk of bias was assessed using the Grading of Recommendations Assessment, Development and Evaluation (GRADE) [32] shown in Appendix A. The avoidable cost PIMs-associated was estimated per patient per year. Costs of the interventions available in different currencies were converted into the 2021 Euro currency. From each reference included in the qualitative analysis, the information extracted is reported in Table 2.

## 3. Results

The search strategy produced 274 potentially relevant publications (Figure 1), of which 16 duplicates were removed. After titles and abstracts screening of the remaining 258 records, we retained 32 potentially relevant publications according to the inclusion criteria. After a full-text review, 12 articles were excluded according to the exclusion criteria. Reasons for exclusion are listed in the diagram (Figure 1). Thus, 20 articles were included in this systematic review.

The 18 included studies were divided according to their study design:(i)Observational studies assessing the frequency of PIPs and associated avoidable costs (*n* = 12);(ii)Trials and observational studies carried out on tailored educational interventions and observations of outcomes in terms of postintervention effectiveness and avoided costs (*n* = 8).

**Figure 1 ijerph-19-06724-f001:**
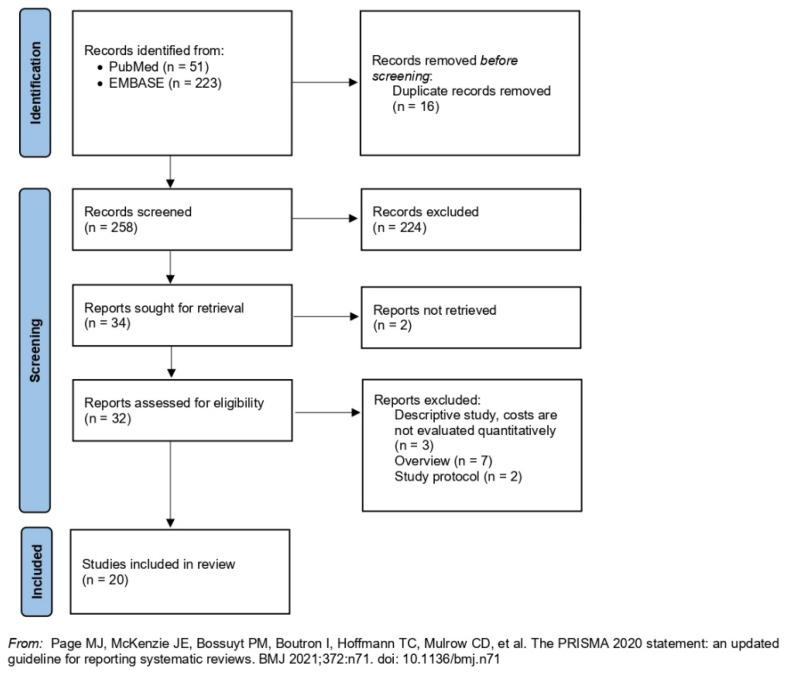
PRISMA diagram of the review’s systematic searches.

### 3.1. Studies without Intervention

Twelve retrospective observational studies were identified, conducted from 2013 to 2020. Table 3 describes the characteristics of these studies. Overall, 60% of the studies reported levels of inappropriate prescribing and associated costs in hospital settings [9,11,12,14,15,16,19,20], while the remaining 40% were conducted among community-dwelling population [10,13,17,18]. In all cases, the target population in which levels of PIPs and costs were investigated consisted of patients over 65 years on a polypharmacy regimen.

Most studies were conducted in America (40%); however, other European (30%) and Asian (30%) countries also investigated frequencies and costs related to PIMs that were identified in different lists. Among all the observational studies, 70% used Beers criteria for the identification of PIMs [9,11,12,14,15,16,19]. Although all studies detected PIMs through the use of different criteria, high rates of inappropriate prescribing were detected. Specifically, in Switzerland, using Beers criteria and the PRISCUS list, the prevalence of PIMs ranged from 21% in community-dwelling population settings [17] to 79% in nursing home settings [12].

Higher avoidable costs were related to higher PIMs prevalence in most cases. In France, a total avoidable healthcare PIP-related expenditure was recorded to be EUR 1449.05 per patient per year [9], in Switzerland, EUR 597.19 per patient per year [12], and in Thailand, it was estimated to be EUR 676.18 per patient per year [11]. Lower costs were detected in Japan [14] and Northern Ireland [19], with a PIMs-related cost of, respectively, EUR 18.66 and EUR 36.71 per patient per year. On the contrary, higher annual PIP-related costs per patient were detected in the US, with an expenditure of EUR 44,258.17 per patient with colorectal cancer [13], and more in general, EUR 11,628.69 per community-dwelling older adults [10].

### 3.2. Studies with Intervention

Eight studies, including an intervention aimed at improving prescribing trends and decreasing costs related to inappropriate medications, were identified. Table 4 describes the characteristics of these studies and their interventions. Studies differed in study design; five were trials or pilot intervention studies [21,22,23,24,25], while the other three were observational studies describing PIPs-related costs as main intervention outcomes [26,27,28]. Overall, 50% of the interventions were carried out in Europe [22,23,26,27]; however, other interventions were performed in the US [25,28], the UK [21], and the Republic of Korea [23]. Most interventions were targeted at clinicians and physicians, particularly general practitioners (GPs). In seven out of eight studies, the intervention included face-to-face meetings with GPs or the medical staff in order to discuss the inappropriate prescriptions to their own assisted patients [21,22,23,24,25,26,28]. Particularly, an intervention conducted in the US was directed at both community pharmacies and ambulatory care clinics, providing specific educational training for pharmacists before the intervention to physicians [25].

Generally, all postintervention outcomes consisted of more appropriate prescribing performance with a reduction in PIPs. On the other hand, 50% of interventions were not cost-effective [21,24,25,26], while 25% of studies recorded a reduction in PIMs-related healthcare costs post intervention between EUR 232.00 and EUR 872.69 per patient per year [22,23], and the remaining 25% studies did not detect a decrease in PIPs-related costs [27,28].

## 4. Discussion

To the best of our knowledge, this is the first systematic review investigating levels and improving the potentially inappropriate medication prescribing (PIP) in older patients and how this improvement may impact avoidable costs to healthcare systems. This review revealed a high prevalence of prescribing inappropriateness at the international level, showing that higher avoidable costs were related to the higher prevalence of potentially inappropriate medication prescriptions (PIM). In the present systematic review, the PIM estimates ranged between 21% and 79%, depending on the explicit criteria used for the assessment and monitoring in heterogeneous healthcare settings.

Corroborating our results, previous systematic reviews have examined associations between PIMs and various outcomes in several settings, which included community settings, nursing homes, hospitals [3,33,34,35,36], and primary care [37,38]. A recent systematic review, carried out in 2021 in the hospital setting [3], confirmed a pooled PIM estimate of 46–56%, depending on the tool used. A second 2021 systematic review carried out in different healthcare settings showed more similar prevalence rates of PIMs ranging from 8.6% in Germany to 81% in Australia [39]. It should be noted that differences in prevalence estimates could be related to the health system of each country, the population considered, and the tools and criteria used to identify PIMs. It is widely recognized that the prevalence estimates of PIMs increase with age and are significantly associated with an increased risk of adverse drug reactions (ADRs) and healthcare costs [9,10,11,12,13,14,15,16,17,18,19,20,22,23].

Important findings of this systematic review confirmed that the consumption of PIMs in older adults has significant economic effects. Although a large body of evidence suggests that the inappropriate use of drugs can impose a high economic burden on society [9,10,11,12,13,14,15,16,17,18,19,20,22,23], to date, there is no concrete estimate of avoidable costs at the European or international level. Several studies evaluated avoidable PIP-related costs at the national level or in a specific setting. From an Irish perspective, total PIP avoidable expenditure was estimated to be about EUR 46 million, showing a significant impact on the national prescribing budget, resulting in 9% of overall yearly expenditure for those aged over 70 years [20]. Our results also revealed that PIM prescription was associated with a higher economic cost in the case of both observational studies monitoring PIP prevalence and consequent outcomes [9,10,11,12,13,14,15,16,17,18,19,20] and studies with an educational intervention [21,22,23,24,25,26,27,28]. From the US perspective, PIM use was significantly associated with greater healthcare utilization and higher healthcare costs in cancer patients, with an avoidable expenditure between EUR 8288.18 and EUR 44,258.17 per patient per year [13]. The same scenario, but detecting significantly lower annual cost per patient, was revealed in a Japanese perspective with an estimate of EUR 18.66–EUR 240.55 avoidable cost per patient [15] by identifying a systematic association between the high prevalence of PIMs and increased healthcare costs.

This systematic review provides a comprehensive exploration of the association between PIPs and a range of health-related outcomes among older adults in heterogeneous settings. Nevertheless, the majority of systematic reviews of the literature to date have only focused on the adverse effects of the PIMs and any associated costs. Accordingly, the main strength of this review is to report for the first time a summary of the worldwide results of educational interventions to improve prescribing appropriateness and the actual effect on avoided healthcare costs. Hence, this study detected that only eight studies [21,22,23,24,25,26,27,28] carried out over the past fifteen years targeted interventions at various healthcare professionals and analyzed the results in terms of avoidable costs. While theory showed that more appropriate prescribing could lead to a decrease in healthcare costs, in practice, most of the interventions conducted to educate healthcare professionals about the rational use of medicine have not proven to be cost effective [21,24]. On the other hand, only two studies, after some tailed interventions carried out in two European settings, recorded a reduction in PIP-related costs. This is the case of French [22] and German [23] interventions directed at healthcare professionals of nursing homes and ambulatory care clinics, respectively, saving about EUR 232.00–EUR 872.69 per patient.

Overall, after most interventions, outcomes consisted of more efficient prescribing trends with a reduction in PIPs, but these interventions were not cost effective [21,22,23]. Various reasons may underlie this evidence. The first one could rely on the information gap between the available scientific evidence and the knowledge integrated by physicians in their decisions and by citizens/patients in their health choices. Secondly, any intervention to reduce inappropriate prescribing should be aware of the presence of financing and incentive logics of companies and professionals based on production, citizens’ and patients’ expectations, health technology turnover, and any conflicts of interest. The implementation science, which aims to improve prescriptive appropriateness, demonstrates that the best results are obtained with multifactorial strategies that combine various interventions in relation to local barriers [40,41]. This should be the starting point for tailoring future interventions to improve prescriptive appropriateness and reduce avoidable healthcare costs.

This review has several limitations that deserve consideration. Firstly, the studies considered used different criteria for the identification and quantification of PIMs, e.g., some studies applied Beers criteria of different versions or STOPP (Screening Tool of Older Persons’ Prescriptions) and START (Screening Tool to Alert to Right Treatment) criteria. These may have caused heterogeneity and variations in estimates. However, in this review, we did not perform a subgroup analysis based on the use of criteria, but compared outcomes in terms of PIM prevalence and its association with health and economic outcomes, regardless of the type of criteria used. Second, it is controversial whether health outcomes are due to PIPs or the disease/condition per se. Comorbidities related to the disease/condition itself might influence negative health outcomes and additional healthcare costs, which is why PIP analysis models should be adjusted for comorbidities. Lastly, regarding the cost estimation of the PIM-related avoidable costs, different currencies were reported by the individual studies. A one-currency conversion was not carried out as the purpose of the review was not cost-quantification but to assess the association between avoidable costs and PIP and the actual effectiveness of health workforce education interventions on healthcare costs.

## 5. Conclusions

This systematic review revealed and confirmed a substantial prevalence of potentially inappropriate prescribing (PIP), which is common among older people. The practice of inappropriate prescribing exists in all care settings and has major clinical and economic consequences. Few educational interventions were carried out over the last fifteen years among different settings to reduce PIP prevalence and avoidable healthcare costs and have resulted in a slowly improving prescribing practice but not cost effectiveness. Implementing cost-effective strategies for reducing the PIP phenomenon and decreasing clinical and economic implications should be an important step in reducing the health systems’ PIP burden.

## Figures and Tables

**Table 1 ijerph-19-06724-t001:** Search strategy.

Query	Keywords (in Mesh/Emtree OR Title and Abstract)	Number of Records	
		Pubmed/Medline	Embase
#1	Inappropriate prescribing OR Appropriate Prescribing OR High-risk medications OR Suboptimal prescribing OR Over-prescribing OR Under-prescribing OR Misprescribing OR Inappropriate Drug OR Inappropriate Medication OR Inappropriate Medicines OR Inappropriate Prescription OR Inappropriate Use OR Medication Appropriateness OR Pharmacological Inappropriateness OR Potential Drug Therapy Problems OR Potentially Harmful Medications OR Prescribing Appropriateness	8946	12,116
#2	Aged OR Aged, 65 and over OR Elderly OR Older Adult OR Older people	3,321,229	2,491,812
#3	Cost OR Cost analysis OR Cost evaluation OR Economic evaluation	594,294	616,13
#4	Intervention OR Action OR General practitioner OR Physician OR Patient	905,693	2,997,717
#1 AND #2 AND #3 AND #4		51	223

**Table 2 ijerph-19-06724-t002:** Data extraction and analysis process: PICO Model.

Data Extraction	Description
Reference	All identification details of the paper
Year	Year of publication
Country	Country in which the study was carried out
Study Design	Type of study conducted
Patient (P)	Population receiving both specific and nonspecific Potentially Inappropriate Prescribing (PIPs)
Intervention (I)	Any type of intervention aimed at health professionals (i.e., physicians, clinicians, pharmacists, nurses), structures (i.e., nursing homes, hospitals, pharmacies), patients, or any monitoring activities of potentially inappropriate prescriptions and costs incurred use any explicit criteria for identifying inappropriateness (such as Beers Criteria, PRISCUS list, STOPP/START criteria)
Comparator (C)	No intervention or any other intervention and no monitoring activities
Outcomes (O)	The postintervention outcome in terms of costs variation and intervention effectiveness; Avoidable costs related to inappropriate prescriptions
Cost types	The perspective of analysis (NHS, society, government, patient) and associated costs (direct healthcare costs, direct non healthcare costs, indirect costs, intangible costs)

**Table 3 ijerph-19-06724-t003:** Characteristics of included descriptive studies without intervention.

Author (Year)	Country	Study Design	Setting	Target Population	Criteria Used	PIPs Frequency	Avoidable Costs (2021 EUR Currency)	Conclusions
Pagès A. et al. (2020) [9]	France	Cross-sectional study	University hospital	Inpatients (*n* = 365)	The EU(7) -PIM List -STOPP/START Criteria	50.4%	EUR 1449.05 per patient per year	1. Substitution of PIPs identified with recommended alternatives was cost saving. 2. Both polypharmacy and type of ward providing care were associated with increased costs of PIMs.
Clark C.M. et al. (2020) [10]	USA	Retrospective cohort study	The 2011–2015 Medical Expenditure Panel Survey (MEPS)	Community-dwelling adults aged > 65 years (*n* = 75,135,061)	2019 AGS Beers Criteria	34.4%	EUR 11,628.69 per patient per year	PIMs continue to be prescribed at a high rate among older adults and are associated with increased costs.
Sattayalertyanyong O. et al. (2020) [11]	Thailand	Prospective study	Medicine wards	Inpatients and outpatients treated with PPIs (*n* = 265)	Guidelines for PPIs	50.6%	EUR 676.18 per patient per year	PPIs are inappropriately prescribed during hospital admission and after discharge, associated with high costs
Rahel S. et al. (2019) [12]	Switzerland	Retrospective cohort study	Nursing homes	Patients aged ≥ 65 years (NHR = 91,166; individuals = 1,364,755)	2015 Beers criteria and the PRISCUS list	79.1%	EUR 597.19 per patient per year	1. Polypharmacy and PIMs are frequent and associated with poor health outcomes in older adults. 2. Drug costs constitute a minor part of the total healthcare costs of these patients.
Feng X. et al. (2019) [13]	USA	Retrospective cohort study	The SEER-Medicare linked database	Older adults with breast (*n* = 17,630), prostate (*n* = 18,721), or colorectal (*n* = 9420) cancer	2015 Beers Criteria	-Breast cancer: 61.7% -Prostate cancer: 47.3% -Colorectal cancer: 66.3%	-Breast cancer: EUR 8288.18 per patient per year -Prostate cancer: EUR 7773.14 per patient per year -Colorectal cancer: EUR 44,258.17 per patient per year	PIMs use was significantly associated with greater healthcare utilization and higher healthcare costs in cancer patients
Tachi T. et al. (2019) [14]	Japan	Retrospective cohort study	Hospital	Inpatients and outpatients aged ≥ 65 years (inpatients = 1236; outpatients = 980)	-Japanese Version (BCJV) -Guidelines for Medical treatment and Its Safety in the Elderly 2015 (GL2015)	-Inpatients BCJV: 24.0%; GL2015: 72.0% -Outpatients BCJV: 26.2%; GL2015: 59.9%	-Inpatients EUR 240.55 per patient per year -Outpatients EUR 18.66 per patient per year	Appropriate use of drugs based on Beers Criteria reduces ADRs and associated costs
Shah K. et al. (2016) [15]	India	Cross-sectional study	Cardiology outpatient department	Patients aged ≥ 65 years (*n* = 236)	2012 Beers criteria	29.3%	EUR 162.76 per patient per year	The high prevalence of PIMs was associated with increased costs in older patients suffering from cardiac diseases
Ladd A.M. et al. (2014) [16]	USA	Retrospective cohort study	Urban hospital	Inpatients and outpatients treated with PPIs (*n* = 2094)	Guidelines for PPIs	76.0%	EUR 2425.34 per patient per year	PPIs are overused in the majority of hospitalized patients with low risk for gastrointestinal bleeding and are associated with high healthcare costs
Blozik E. et al. (2013) [17]	Switzerland	Retrospective cohort study	Community-dwelling population	Beneficiaries of health service (*n* = 5000)	-2003 Beers criteria -PRISCUS list.	21.0%	EUR 1861.77 per patient per year	1. The prevalence of polypharmacy and PIMs in the adult and elderly was high; 2. The elderly were associated with higher costs
Dionne P.-A. et al. (2013) [18]	Canada	Retrospective cohort study	Community-dwelling population	Beneficiaries of health service aged ≥ 65 years (*n* = 744)	2003 Beers criteria	44.0%	EUR 2567.67 per patient per year	A significant association between benzodiazepine-related drug interactions and healthcare costs.
Bradley M.C. et al. (2012) [19]	Northern Ireland	Cross-sectional study	Hospital	Patients aged ≥ 70 years (*n* = 166,108)	-STOPP criteria -Beers criteria	34%	EUR 36.71 per patient per year	The prevalence of PIP was high among the study cohort, increased with polypharmacy, and was associated with a significant cost.
Cahir C. et al. (2010) [20]	Ireland	Retrospective national population study	Geriatric units, nursing homes and hospitals	Patients aged ≥ 70 years (*n* = 338,801)	2007 STOPP criteria	36%	EUR 134.68 per patient per year	The findings identify a high prevalence of PIP in Ireland with significant cost consequences.

Abbreviations: EU(7)-PIM List: European Union (7)-potentially inappropriate medication; PIM: potentially inappropriate medication; PIP: potentially inappropriate prescribing; PPI: proton pump inhibitors; STOPP/START Criteria: Screening Tool of Older Persons’ Prescriptions and Screening Tool to Alert to Right Treatment Criteria.

**Table 4 ijerph-19-06724-t004:** Characteristics of included studies with intervention.

Author (Year)	Country	Study Design	Intervention Aim	Time Frame	Setting	Target Population	Intervention	Outcome	Avoidable Costs (2021 EUR Currency)	Conclusions
**Trials**										
Desborough J.A. et al. (2020) [21]	England	Cluster randomized controlled trial	To determine the clinical cost-effectiveness of multiprofessional medication review service (MPMR).	1 year	Care homes	Care home medical staff (*n* = 826)	Intervention care homes received an MPMR from a team consisting of a clinical pharmacist, GP, and a care home member.	1. Intervention reduced PIMs by 20% at 12 months 2. Intervention group had higher costs and falls per person per year.	/	The intervention was dominated by usual care and would not be considered cost-effective.
Leguelinel-Blache G. et al. (2020) [22]	France	Monocentric before-after pilot and paired study	To assess the impact of multidisciplinary medication review (MMR) and costs incurred by the hospital and the national health service.	1 year	Nursing homes	-Nurses -GPs (*n* = 49)	Two hospital pharmacists, using different criteria, reviewed patients’ prescriptions and conducted multidisciplinary meetings suggesting modifications to the patients’ medical team.	The number of patients taking at least one PIMs decreased from 30.6% before to 6.1% after the intervention.	EUR 232.00 per patient per year	The MMR reduced the iatrogenic drug risk for elderly residents and costs from the nursing home perspective, particularly drug expenditure.
Whitman A. et al. (2018) [23]	Germany	Pilot study	1. To compare the application of three geriatric medication screening tools to the Beers Criteria alone for PIM quantification 2. To determine the feasibility of a pharmacist-led polypharmacy assessment.	9 months	Ambulatory care clinic	-Geriatric oncologist -Patient -Caregiver (*n* = 26)	1. Pharmacist performed an assessment of all drug therapies by reviewing all PIPs through different criteria 2. Reduction in prescribing occurred after a discussion with a pharmacist, oncologist, patient, or caregiver.	After the application of the three-tool assessment, 73% of PIMs identified were deprescribed, resulting in a mean of 3 medications deprescribed per patient.	EUR 872.69 per patient per year	1. The three-tool assessment identified 3 times more PIMs than the Beers Criteria alone. 2. Pharmacist-led deprescribing interventions were feasible, leading to improved patient outcomes and cost savings.
Kim S.J. et al. (2018) [24]	Republic of Korea	Interrupted time-series study design	To evaluate the effect of the prospective drug utilization review (DUR) system to improve prescribing practices, adverse drug events (ADEs), and healthcare expenditure.	Rolling 6-year period	Outpatient	Patients with musculoskeletal or connective tissue disorders (*n* = 54,58)	Introduction of DUR systems for monitoring drugs’ prescription operating prospectively and retrospectively, providing feedback to the provider.	More efficient prescribing, reduction in DDIs, and increase in the use of gastro-protective drugs.	/	The intervention had a positive effect on patient outcomes but was not associated with reduced ADE-related costs.
Christensen D.B. et al. (2007) [25]	USA	Before/after design with two control groups	To assess the feasibility of a pharmacist-based Medication Therapy Management (MTM) service.	8 months	-Community pharmacies -Ambulatory care clinic	-Community and Ambulatory care Pharmacists -Volunteering patients (*n* = 1000)	1. Educational training for pharmacists 2. MTM-type program offered to patients with polytherapy	1. Pharmacists identified an average of 3.6 potential drug therapy problems (PDTPs) per patient at the first visit. 2. Pharmacists recommended a drug therapy change in about 50% of patients and contacted the prescriber more than 85% of the time. 3. No significant differences were observed in patient co-payment or insurer prescription costs.	/	1. The intervention reduced the number of potential drug therapy problems 2. The intervention did not necessarily result in reductions in prescription drug use or cost.
**Observational studies**										
Foubert K. et al. (2020) [26]	Belgium	Prospective observational study	To investigate the acceptance of pharmacist recommendations based on a screening tool for PIP: Ghent Older People’s Prescriptions community Pharmacy Screening (GheOP3S)-tool.	5 months	Nursing homes	-Pharmacists -GPs -Nurse (*n* = 50)	1. Collection of the medication list for each patient 2. Lists’ screening using the GheOP3 S-tool and formulation of recommendations for every detected GheOP3 S-criterion 3. Face-to-face pharmacist-GP meetings to discuss the pharmacist recommendations, resulting in an agreed action plan 4. Final meeting between the pharmacist, head nurses, and coordinating physician to communicate these plans.	1. Most pharmacist recommendations on PIP considered stopping the medication 2. The 45% of relevant recommendations were accepted by the GPs 3. Number of GheOP3S-criteria and medication costs remained unchanged	/	The acceptance and implementation of pharmacist recommendations were relatively low
Fischer K.E. et al. (2018) [27]	Germany	Prospective observational study	To analyze costs and quality of prescribing conditional on the level of utilization of the drug budget	7 years	Outpatient	Physicians (*n* = 440)	drug-budgets introduction and motoring of Drug Budget for Physicians, the level of drug budget utilization, and differentiation by varying levels of enforcement where physicians overspent their budgets.	The level of drug budget utilization influences the cost and quality of prescribing PIMs to the elderly.	/	1. Drug use expressed as the number of prescriptions per visit had not changed 2. The cost of prescribing changed when a drug budget mechanism was put in place
Reeve E. et al. (2015) [28]	Australia	Prospective feasibility study	To assess the feasibility of a patient-centered deprescribing process	6 months	Hospital outpatient clinics	-GPs -Patients (*n* = 43)	1. Identification of PPIs by Pharmacists 2. Determine if the medication can be discontinued by GPs 3. Withdrawal of PPI.	Of the eight participants who were invited to have their PPI withdrawn, six were willing to undergo trial withdrawal, and all achieved cessation/dose reduction.	/	1. The patient-centered deprescribing process can safely reduce inappropriate PPI prescribing 2. Cost-effectiveness of this approach needs to be determined

Abbreviations: PIM—potentially inappropriate medication; PIP—potentially inappropriate prescribing; GP—General practitioner.

## Data Availability

Not applicable.

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
