# Peer review of "The Effectiveness of Interventions to Evaluate and Reduce Healthcare Costs of Potentially Inappropriate Prescriptions among the Older Adults: A Systematic Review"

_ijerph, 2022, doi:10.3390/ijerph19116724_

Round 1

Reviewer 1 Report

In this systematic review the authors aimed to evaluate the impact of potentially inappropriate prescribing (PIP) on avoidable costs and of PIP interventions on costs. However, the paper presents serious flaws from the methodological point of view that need to be addressed at least with a major revision.

First of all, even if stating that the paper adheres to the PRISMA reporting guidelines, many important pieces are missing (e.g. protocol availability, registration, torough description of inclusion criteria etc). I would suggest to review the PRISMA guidelines and checklist.

Most importantly, the authors should better define the research question(s): if focusing on the impact of interventions on costs why include cross sectional studies and studies not reporting costs as the outcome? This reflects on poor quality of inclusion and exclusion criteria definition.

The search strategy is too simple and this could result in loss of significant papers. I would suggest the authors to refer to a specialized librarian to create a sound search strategy. For example, use of jolly characters (e.g. prescri*) and inclusion of relevant synonyms (e.g. "geriatric patients") would be of use. Also, the inclusion in the search strategy of the "intervention" set of keywords probably narrows research too much not providing specificity to the search results. Moreover the authors did not balance these issues by consulting relevant bibliographies of selected studies and did not consult grey literature sources and clinical trial registries, that can provide much relevan information. Lastly, search strategies need to be re-run since in a 1 year time period relevant studies probably could be published.

In the methods section, how authors screened titles/abstract, full texts and extracted information is ill-described (e.g. how many? they were blind?). Moreover, risk of bias assessment is described in the methods section but not reported in the results and discussion. Also, an evaluation of publication bias is mandatory given the field.

The tables are well constructed but I believe that they included not relevant or not scientifically sound information (e.g. many "conclusions" cannot be drawn from study design and results) and omit potentially relevant information, including: number of patients enrolled and basic characteristics (age, sex, conditions), population and setting should be better defined. In the intervention table, also intervention needs to be better defined (including type of criteria used...) and outcomes need to be reported with more detail (e.g. OR, p values, % of PIPs etc).

I would also suggest standardizing costs in terms of currency and of timespan (e.g. USD/EUR per patient per day).

Please avoid the use of the term elderly since it can be perceived as discriminatory and prefer the term "older adults/subjects/patients". Also, avoid using the lLatin term "die" since it can be confused with the English verb "to die".

Author Response

Response to Reviewer 1 Comments

Point 1: In this systematic review the authors aimed to evaluate the impact of potentially inappropriate prescribing (PIP) on avoidable costs and of PIP interventions on costs. However, the paper presents serious flaws from the methodological point of view that need to be addressed at least with a major revision.

Response 1: Dear reviewer, we would like to thank you for the useful inputs. We addressed some of your suggestions as follow.

Point 2: First of all, even if stating that the paper adheres to the PRISMA reporting guidelines, many important pieces are missing (e.g. protocol availability, registration, torough description of inclusion criteria etc). I would suggest to review the PRISMA guidelines and checklist.

Response 2: We reviewed the PRISMA guidelines and checklist and modify what needed. We better defined the inclusion criteria of the systematic review. Previously, the study protocol was already developed beforeahed to start the review, but, as this review is part of a broader national project aimed to improve prescripion appropriateness in older adults. The tight timeline for project closure directed us to begin the review immediately. As you know, study protocols for reviews already in the data analysis phase cannot be registered. In any case, we checked that duplicates of this review were not present.

Point 3: Most importantly, the authors should better define the research question(s): if focusing on the impact of interventions on costs why include cross sectional studies and studies not reporting costs as the outcome? This reflects on poor quality of inclusion and exclusion criteria definition.

Response 3: We understand your point of view, reason why we better defined inclusion criteria.

Point 4: The search strategy is too simple and this could result in loss of significant papers. I would suggest the authors to refer to a specialized librarian to create a sound search strategy. For example, use of jolly characters (e.g. prescri*) and inclusion of relevant synonyms (e.g. "geriatric patients") would be of use.

Response 4: We understand your concerns. The research strategy was intended to be intentionally simple in order to identify, without undue restraint, the state of the art regarding actions to improve prescriptive appropriateness, if any, and the impact in economic terms on health systems assessed to date.

Point 5: Also, the inclusion in the search strategy of the "intervention" set of keywords probably narrows research too much not providing specificity to the search results.

Response 5: The use of the term 'intervention' and 'action' was included in the search strategy precisely to identify any trials that aimed to improve prescriptive appropriateness. These terms, however, did not have a restrictive meaning in the search strategy, they were included, in fact, with the Boolean operator OR (see below). This is the reason why many records included did not include specific interventions but observational studies.

((((((Intervention[MeSH Terms]) OR (action[MeSH Terms])) OR (general practitioner[MeSH Terms])) OR (clinician[MeSH Terms])) OR (physician[MeSH Terms])) OR (patient[MeSH Terms]))

Point 6: Moreover the authors did not balance these issues by consulting relevant bibliographies of selected studies and did not consult grey literature sources and clinical trial registries, that can provide much relevan information.

Response 6: This aspect was defined by the principle of the analysis. By choice, we decided to dwell only on a strategy that includes indexed bibliographic databases (PubMed/Medline and Embase) and not investigate the gray literature.

Point 7: Lastly, search strategies need to be re-run since in a 1 year time period relevant studies probably could be published.

Response 7: We really undertand this aspect. As mentioned earlier, this review is part of a larger national project. For this reason, the need arisen from this study was to evaluate any interventions to improve prescriptive appropriateness carried out up to February 2021. Certainly after this study, we will update the analysis to date.

Point 8: In the methods section, how authors screened titles/abstract, full texts and extracted information is ill-described (e.g. how many? they were blind?).

Response 8: Thanks for suggestion. We added this information in the methods section.

Point 7: Also, an evaluation of publication bias is mandatory given the field.

Response 9: Please provide your response for Point 1. (in red)

Point 9: The tables are well constructed but I believe that they included not relevant or not scientifically sound information (e.g. many "conclusions" cannot be drawn from study design and results) and omit potentially relevant information, including: number of patients enrolled and basic characteristics (age, sex, conditions), population and setting should be better defined. In the intervention table, also intervention needs to be better defined (including type of criteria used...) and outcomes need to be reported with more detail (e.g. OR, p values, % of PIPs etc).

Response 10: We implemented the tables with adjustment information found in the papers, such as the number of subjects included in the studies.

Point 10: I would also suggest standardizing costs in terms of currency and of timespan (e.g. USD/EUR per patient per day).

Response 11: We’ve adapted and converted the currency and the timespan (EUR per patient per year).

Point 7: Please avoid the use of the term elderly since it can be perceived as discriminatory and prefer the term "older adults/subjects/patients". Also, avoid using the lLatin term "die" since it can be confused with the English verb "to die"..

Response 11: We have changed the term "elderly" to "older adults" and the term "die" to "per day" throughout the manuscript.

Reviewer 2 Report

Dear Authors,

The manuscript ID: ijerph-1658294 entitled “The effectiveness of interventions to evaluate and reduce healthcare costs of potentially inappropriate prescriptions: a systematic review” written by Sara Mucherino, Manuela Casula, Federica Galimberti, Ilaria Guarino, Elena Olmastroni, Elena Tragni, Valentina Orlando, Enrica Menditto (on behalf of the EDU.RE.DRUG Group) is very interesting.

In my opinion, the purpose of this review – explore the state of the art regarding potentially inappropriate prescriptions (PIP) related avoidable costs and results of any interventions to reduce PIP and the impact on healthcare costs is very interesting and topical. The whole manuscript (Introduction, Materials and Methods, Results, Discussion, Conclusions) is properly organized. Introduction is concise and contains general data on population ageing and potentially inappropriate prescribing, which is associated with increased risk of adverse drug reactions and increased healthcare costs. A systematic review of the literature was conducted according to the PRISMA statement guidelines. An appropriate search strategy was used to prepare this review. The results are documented, summarized in the form of figure or tables and correctly interpreted. Based on the results, adequate discussion and conclusions were drawn that the prevalence of potentially inappropriate prescribing in the elderly is common and significant. Moreover, the practice of inappropriate prescribing exists in all care settings and has major clinical, economic consequences.

It is a well written and original review. According to me, this manuscript is very valuable and may be accepted for the publication in “International Journal of Environmental Research and Public Health”.

With highest regards,

Author Response

Point 1: In my opinion, the purpose of this review – explore the state of the art regarding potentially inappropriate prescriptions (PIP) related avoidable costs and results of any interventions to reduce PIP and the impact on healthcare costs is very interesting and topical. The whole manuscript (Introduction, Materials and Methods, Results, Discussion, Conclusions) is properly organized. Introduction is concise and contains general data on population ageing and potentially inappropriate prescribing, which is associated with increased risk of adverse drug reactions and increased healthcare costs. A systematic review of the literature was conducted according to the PRISMA statement guidelines. An appropriate search strategy was used to prepare this review. The results are documented, summarized in the form of figure or tables and correctly interpreted. Based on the results, adequate discussion and conclusions were drawn that the prevalence of potentially inappropriate prescribing in the elderly is common and significant. Moreover, the practice of inappropriate prescribing exists in all care settings and has major clinical, economic consequences.

It is a well written and original review. According to me, this manuscript is very valuable and may be accepted for the publication in “International Journal of Environmental Research and Public Health”.

Response 1: Dear Reviewer, many thanks for your positive consideration and for stressing the main goal of our work. On behalf of all the co-authors, we sincerely appreciate your interest in this study. We agree that the topic of the correct use of medicines is a crucial one in clinical practice, and, strongly consider that implementation of cost-effective strategies is crucial to reduce potentially inappropriate prescriptions. Therefore, in the next future research on cost-effective strategies needs to be implemented in order to improve prescribing and move forward.

Reviewer 3 Report

The research topic is interesting and timely. Most importantly, while there are plenty of research on potentially inappropriate prescribing in older patients much fewer address evaluation of its costs to healthcare systems. Thus, providing systematic review of such research is very needed. Moreover, the review itself was well designed and the description of methodology is appropriate and very clear. I have only few minor comments, which, I hope, the Authors, may find useful:

  1. Neither the tittle nor the study aim (lines 64-66) refer to the elderly population. Of course among the inclusion criteria they list “ii) elderly population” (line 84), however it could be added in the title to avoid confusion. Especially that in the Discussion section (line 191) they state explicitly: “this is the first systematic review investigating levels and/or improving potentially inappropriate prescribing (PIP) in older patients”. Otherwise the readers may wonder why the Authors do not mention potentially inappropriate prescribing in children.
  2. Abstract, line 17: it seems that it should be “was to explore and compare” not “was to explore compare”
  3. While the authors list abbreviations under each table I would also recommend listing of all abbreviations at the end of the main text among other sections (Funding, Author Contributions or Conflict of Interest). Especially, that some abbreviations that do appear in the text are not listed, i.e. ADRs or AIFA.
  4. line 60: abbreviation of the Italian Medicine Agency: AIFA should be preceded by its original name: Agenzia Italiana del Farmaco – AIFA.
  5. Although the Authors analyse the most important research on the PIP among the elderly there is at least one studies they have missed, i.e.:

-- Bradley MC, Fahey T, Cahir C, Bennett K, O'Reilly D, Parsons C, Hughes CM. Potentially inappropriate prescribing and cost outcomes for older people: a cross-sectional study using the Northern Ireland Enhanced Prescribing Database. Eur J Clin Pharmacol. 2012 Oct;68(10):1425-33. doi: 10.1007/s00228-012-1249-y.

Regardless of those minor remarks I appreciate this research a lot. On the whole, I find the topic original, interesting and timely and recommend its publication in IJERPH. I am convinced that the issues raised in the article will stimulate the discussion on the potentially inappropriate prescribing (PIP) in older patients.

Author Response

Point 1: The research topic is interesting and timely. Most importantly, while there are plenty of research on potentially inappropriate prescribing in older patients much fewer address evaluation of its costs to healthcare systems. Thus, providing systematic review of such research is very needed. Moreover, the review itself was well designed and the description of methodology is appropriate and very clear. I have only few minor comments, which, I hope, the Authors, may find useful.

Response 1: Dear Reviewer, thanks for stressing the main goal of the study. We found you comments very useful and implemented in the main text.

Point 2: Neither the tittle nor the study aim (lines 64-66) refer to the elderly population. Of course among the inclusion criteria they list “ii) elderly population” (line 84), however it could be added in the title to avoid confusion. Especially that in the Discussion section (line 191) they state explicitly: “this is the first systematic review investigating levels and/or improving potentially inappropriate prescribing (PIP) in older patients”. Otherwise the readers may wonder why the Authors do not mention potentially inappropriate prescribing in children.

Response 2: We agree with your observation. Therefore, we have implemented the title and aim of the study with the specific population information. The title has been modified as follows: "The effectiveness of interventions to evaluate and reduce healthcare costs of potentially inappropriate prescriptions among the older adults: a systematic review"

Point 3: Abstract, line 17: it seems that it should be “was to explore and compare” not “was to explore compare”

Response 3: We modified the typo as follows “Study aim was to explore and compare results (..)”.

Point 4: While the authors list abbreviations under each table I would also recommend listing of all abbreviations at the end of the main text among other sections (Funding, Author Contributions or Conflict of Interest). Especially, that some abbreviations that do appear in the text are not listed, i.e. ADRs or AIFA.

Response 4: Thanks for the suggestion. We went through the whole manuscript and made all the abbreviations explicit especially when they were first mentioned.

Point 5: line 60: abbreviation of the Italian Medicine Agency: AIFA should be preceded by its original name: Agenzia Italiana del Farmaco – AIFA.

Response 5: We have specified also the original name.

Point 6: Although the Authors analyse the most important research on the PIP among the elderly there is at least one studies they have missed, i.e.:

-- Bradley MC, Fahey T, Cahir C, Bennett K, O'Reilly D, Parsons C, Hughes CM. Potentially inappropriate prescribing and cost outcomes for older people: a cross-sectional study using the Northern Ireland Enhanced Prescribing Database. Eur J Clin Pharmacol. 2012 Oct;68(10):1425-33. doi: 10.1007/s00228-012-1249-y.

Response 6: Thanks for the useful suggestion. The study was included in the discussion by was missed in the table. We provided to add it in the analysis.

Point 7: Regardless of those minor remarks I appreciate this research a lot. On the whole, I find the topic original, interesting and timely and recommend its publication in IJERPH. I am convinced that the issues raised in the article will stimulate the discussion on the potentially inappropriate prescribing (PIP) in older patients.

Response 7: On behalf of all the co-authors, we thank you for your positive consideration.

Reviewer 4 Report

Thank you for the manuscript. It was an interesting read, and in a very important field as well. Given the paucity of information available, getting a systematic review of the literature helps to set a tone for future discussion in this topic. The manuscript is acceptable for publication, however, does have a few caveats attributed to it.

Major comments

  1. English-editing is necessary to ensure the readability of the manuscript increases, as there are several grammatical issues at present;
  2. In the introduction, patient communication is listed as a primary issue in PIP, however, there are other general concerns as well, which include non-compliance, seeking non-medical practitioner treatment, complementary and alternative treatment, or even medical practitioner competency concerns (which is also described heavily to contribute to drug duplication, irrational prescription, and so forth). This needs to be acknowledged to present a more holistic view, and also aligns to many of the intervention aspects discussed; and
  3. Elderly populations are listed as one of the pillars of the strategy, however, the age range starts at 40, which according to literature would be younger than some accepted elderly ranges of 65 years (Petry N. The Gerontologist 2002;42:92-99). This should be clarified or amended.

Minor comments

  1. Please justify the selection of the databases used;
  2. Please elaborate on the various criteria used, such as Beers, PRISCUS, and so forth
  3. Using the specific currency gives context, however, makes it difficult for the reader to see what the difference may be beyond that. I recommend selecting a singular currency for conversion in a separate column of the tables to showcase comparative strength;
  4. Some of the interventions are a bit vague in their description, and thus its unclear what type of medicine review, training, and/or communications were held. This will help support why some are more feasible than others; and
  5. Given the competency issues often seen with such prescription issues, I would recommend including a potential curricular review of medical programmes as a general statement as this would help treat the underlying cause for such problems as well, and may thus help support additional interventions later on.

Author Response

Point 1: Thank you for the manuscript. It was an interesting read, and in a very important field as well. Given the paucity of information available, getting a systematic review of the literature helps to set a tone for future discussion in this topic. The manuscript is acceptable for publication, however, does have a few caveats attributed to it.

Response 1: Dear Reviewer, thanks for your pertinent consideration. Indeed, we believe that a synthesis of the improvement interventions conducted so far, albeit few, can raise awareness towards the design of further ad-hoc interventions for the improvement of prescriptive appropriateness.

Point 2: Major comments: English-editing is necessary to ensure the readability of the manuscript increases, as there are several grammatical issues at present;

Response 2: Thank you for your suggestion. We reviewed the entire manuscript.

Point 3: Major comments: In the introduction, patient communication is listed as a primary issue in PIP, however, there are other general concerns as well, which include non-compliance, seeking non-medical practitioner treatment, complementary and alternative treatment, or even medical practitioner competency concerns (which is also described heavily to contribute to drug duplication, irrational prescription, and so forth). This needs to be acknowledged to present a more holistic view, and also aligns to many of the intervention aspects discussed;

Response 3: We agree with your considerations, reason why we implemented the introduction by better explaining potential reasons of inappropriate prescribing as follows: “In this scenario the lack of appropriate dialogue between doctor -patient and doctors competency concerns patient may result in inappropriate prescribing as prescribers may lack important information justifying the prescription of a particular drug or therapy [7]. Thus, prescriptive inappropriateness can result from multiple pa-tient-related reasons, such as seeking nonmedical treatment, complementary and al-ternative treatment, not in agreement with the prescriber. Another risk factor for pre-scriptive inappropriateness is age. In fact, with advancing age, the complexity of the patient's clinical picture increases, which can result in inappropriate polypharmacy and poor adherence to drug therapies.”

Point 4: Major comments: Elderly populations are listed as one of the pillars of the strategy, however, the age range starts at 40, which according to literature would be younger than some accepted elderly ranges of 65 years (Petry N. The Gerontologist 2002;42:92-99). This should be clarified or amended.

Response 4: This study is part of a larger project funded by the Italian Medicine Agency (AIFA) titled EDU.RE.DRUG Project (Effectiveness of informative and/or educational interventions aimed at improving the appropriate use of drugs designed for general practitioners and their patients). In whole, the project analysed informations on doctors and their patients over 40 years to have an overall scenario and to evaluate difference in prescribing trends after educational intervensions. In the present study we only focused on the interventions carried out among the elderly population (≥65 years). We rephrased the sentence “The EDU.RE.DRUG Project aims to evaluate the appropriateness of drug prescription in people aged ≥40 years living in Northern and Southern Italy [27,28].” to avoid any misunderstanding.

Point 5: Minor comments: Please justify the selection of the databases used;

Response 5: We justified the use of MEDLINE/Pubmed and Embase Databases as follow: “Although MEDLINE was chosen as the source as the main bibliographic database containing more than 29 million references to journal articles in the life sciences, PubMed/MEDLINE does not fully cover the research literature in the field. For this reason, Embase was used as a supplementary source to Medline/PubMed to get a comprehensive overview of the papers in the field of potentially inappropriate prescribing and related avoidable costs.”

Point 6: Minor comments: Please elaborate on the various criteria used, such as Beers, PRISCUS, and so forth

Response 6: We added a specification more in the introduction section as follow: “Among criteria most used to date were: the Beers Criteria for Potentially Inappropriate Medication Use in Older Adults, commonly called the Beers List, which are guidelines to improve prescribing appropriateness in older adults (65 years and older); The PRISCUS list, firstly created for the German pharmaceuticals market and also developed to improve prescribing appropriateness in older adults; STOPP (Screening Tool of Older Persons' Prescriptions) and START (Screening Tool to Alert to Right Treatment) criteria, developed for clinicians and aimed to review prescription inappropriateness in older adults.”

Point 7: Minor comments: Using the specific currency gives context, however, makes it difficult for the reader to see what the difference may be beyond that. I recommend selecting a singular currency for conversion in a separate column of the tables to showcase comparative strength;

Response 7: We provided to made a conversion in the same currency and timespan (EUR per patient per year).

Point 8: Minor comments: Some of the interventions are a bit vague in their description, and thus its unclear what type of medicine review, training, and/or communications were held. This will help support why some are more feasible than others;

Response 8: We provided to explain more in depht interventions not detailed.

Point 9: Minor comments: Given the competency issues often seen with such prescription issues, I would recommend including a potential curricular review of medical programmes as a general statement as this would help treat the underlying cause for such problems as well, and may thus help support additional interventions later on.

Response 9: Many thanks for the valuable input. This is definitely something discussed among the authors, as well as a research action that will follow this study.

Round 2

Reviewer 4 Report

The manuscript has been improved quite a bit. Thank you for the explanations provided as well. Only the following elements have been picked up for clarification:

1. Line 46: Although this helps to expand on other forms of irrational prescribing, starting off with 'In this scenario' creates some ambiguity. I suggest these sentences be rewritten to acknowledge the multifactorial nature of the problem very briefly, and then mention that the focus will be on communicative competencies.

Author Response

Response to Reviewer 4 Comments

Point 1: The manuscript has been improved quite a bit. Thank you for the explanations provided as well. Only the following elements have been picked up for clarification:

  1. Line 46: Although this helps to expand on other forms of irrational prescribing, starting off with 'In this scenario' creates some ambiguity. I suggest these sentences be rewritten to acknowledge the multifactorial nature of the problem very briefly, and then mention that the focus will be on communicative competencies.

Response 1: Dear Reviewer, many thanks for the advice. We modified as suggested.